# Uncertainties in critical slowing down indicators of observation-based fingerprints of the Atlantic Overturning Circulation

**Maya Ben-Yami** [1,2] ✉, **Vanessa Skiba** [2], **Sebastian Bathiany** [1,2] & **Niklas Boers** [1,2,3]

Observations are increasingly used to detect critical slowing down (CSD) to measure stability changes in key Earth system components. However, most datasets have non-stationary missing-data distributions, biases and uncertainties. Here we show that, together with the pre-processing steps used to deal with them, these can bias the CSD analysis. We present an uncertainty quantification method to address such issues. We show how to propagate uncertainties provided with the datasets to the CSD analysis and develop conservative, surrogate-based significance tests on the CSD indicators. We apply our method to three observational sea-surface temperature and salinity datasets and to fingerprints of the Atlantic Meridional Overturning Circulation derived from them. We find that the properties of these datasets and especially the specific gap filling procedures can in some cases indeed cause false indication of CSD. However, CSD indicators in the North Atlantic are still present and significant when accounting for dataset uncertainties and non-stationary observational coverage.

In recent years there has been increasing focus on non-linearities and the potential of abrupt transitions in the Earth system, especially in response to anthropogenic greenhouse gas emissions. Of particular interest are systems that have multiple stable equilibrium states, and so could rapidly transition in a self-perpetuating way to a different state once a critical forcing threshold is reached[1]. When such systems approach a transition to a different state in response to gradual changes in forcing, they may exhibit so-called critical slowing down (CSD), in which their response to perturbations changes in a characteristic manner[2]. CSD can be a sign of a forthcoming transition and may in certain situations be used to anticipate it; statistical signs of CSD, such as increasing variance or autocorrelation, have hence also been termed early-warning signals[3]. CSD has been identified in observations of numerous Earth system components that have been identified as tipping elements[1]. These include the Greenland Ice Sheet[4], the Atlantic Meridional Overturning Circulation (AMOC)[5,6], the Amazon rainforest[7], as well as other parts of global vegetation[8,9].

However, CSD indicators such as the variance and lag-one autocorrelation[3] or the restoring rate[5,10] are calculated from observational datasets that are not optimised to capture higher-order statistics. Observational datasets employ a variety of methods to combine data from different instruments, adjust observational biases and fill in missing grid cells. These methods are tuned to best capture mean global trends in the data, sometimes at the cost of underlying statistical properties. For example, variance may decrease just as a result of increasing data coverage. This can be caused by an increasing numbers of observations that can be taken into account for each reported value, e.g. by taking the mean over samples of increasing size and correspondingly reduced standard error. However, this is just a simplified example, and as each observational dataset has its own specific methods of data assimilation and infilling, it is not possible to generalize the effect that observational dataset uncertainties would have on higher-order statistics. For an in-depth investigation of data aggregation effects using remote sensing data we refer to Smith et al.[8].

[1]Earth System Modelling, School of Engineering and Design, Technical University of Munich, Munich, Germany. [2]Potsdam Institute for Climate Impact Research, Potsdam, Germany. [3]Department of Mathematics and Global Systems Institute, University of Exeter, Exeter, UK. ✉e-mail: maya.ben-yami@tum.de

In this work we focus on sea-surface temperature (SST) and salinity based CSD indicators for the AMOC. We show how dataset uncertainties can be incorporated into the CSD analysis, and how the standard significance testing methods can be modified to account for the influence of different infilling methods.

This study is based on the work by Boers et al. 2021[5] (hereafter B21). B21 analysed SST- and salinity-based proxies of the AMOC strength[11,12] to investigate whether a declining stability of the AMOC can be detected from statistical indicators. The AMOC is a key element of the Earth's climate system, transporting large amounts of heat and salt northward in the upper layers of the Atlantic Ocean. Paleoclimate proxy evidence as well as theoretical considerations suggest that the AMOC is bistable, with a second, substantially weaker circulation mode in addition to the present strong mode[13–18]. The bistability of the AMOC has recently been supported by comprehensive high-resolution model simulations[19]. There are several lines of proxy- and observation-based evidence suggesting that the AMOC has indeed weakened in the last decades to centuries[20], although the decline and its cause are still controversial[21,22]. Finally, comprehensive models predict that the AMOC will weaken further under anthropogenic global warming[23].

The most commonly used CSD indicators are an increase in the variance and autocorrelation of a time series. However, these indicators can result in false positives, as an increase in the variance or autocorrelation can also be caused by a corresponding statistical change in the external conditions, such as an increase in the autocorrelation of the driving noise. To avoid such false positives, B21 introduced the corrected restoring rate, estimated under the assumption of non-stationary correlated noise driving the system. We simply refer to as restoring rate $\lambda$ in the following. For a system in state $x$ close to equilibrium, we can linearize about the equilibrium state and thus the dynamics can be approximated as $\frac{dx}{dt} \approx \lambda x + \eta$, where $\eta$ stands for random external perturbations. $\lambda$ can thus be estimated by regressing and estimating of the derivative $dx/dt$ against $x$. One can then avoid false CSD indicators caused by the properties of $\eta$ by performing this regression with a generalized least square algorithm that assumes noise with varying autocorrelation (for more details see B21). $\lambda$ is negative for systems close to a stable state, and when a multistable system approaches a critical transition, $\lambda$ increases to 0 from below. We focus on $\lambda$ in the main text of the paper; corresponding analyses and figures for the variance and autocorrelation can be found in the Supplementary Materials.

A prerequisite for a statistically significant increase in CSD indicators is a sufficiently long time series. Direct observations of the AMOC strength in the Northern Atlantic only go back to 2004[24]. Consequently, numerous AMOC fingerprints based on observations spanning longer time periods have been suggested, which are thought to reflect variations in the strength of the AMOC. As the AMOC transports heat and salt northward, SSTs and salinity profiles are commonly used as AMOC fingerprints. B21 took two approaches to identifying CSD indicators for the AMOC. The first is to look for CSD indicators in previously identified fingerprints that are constructed by averaging SST or salinity over a specific region[11,12,25,26]. For example, one such fingerprint was proposed by Caesar et al. 2018[12] and is calculated by taking the average SST in the subpolar gyre region minus the average global SST (see also[11]). The second approach is to calculate the CSD indicators for each grid cell in the SST or salinity dataset, and look at the regions that are thought to be related to the AMOC strength. For example, if the AMOC weakens, salinity is accumulated along its main transport path, and thus the changes in near-surface salinity along the Gulf Stream and North Atlantic Current are thought to reflect changes in the strength of the AMOC[15]. B21 found significant increases in $\lambda$ both in the SST and salinity fingerprints and on spatially explicit maps, and both of these approaches will be used in this work.

B21 used three observational datasets: the HadISST1[27] and ERSSTv5[28] datasets for SSTs and the EN4.2.1 dataset[29] for salinity

profiles. They provide smooth global fields from 1871 and 1854 to present for the SST data, respectively, and from 1900 to present for the salinity data. In this work we use an updated version of the EN4.2.2 salinity data (EN4) and for further robustness testing additionally use the HadSST4[30] and HadCRUT5[31] SST datasets, which date back to 1850.

Although this study will focus on SST and salinity datasets and on CSD indicators, the work presented here can be generalised to many other datasets and higher-order statistics (for example[32]).

## Results

### Uncertainty ensembles

In this section, we asses how uncertainties provided with the observational datasets propagate to uncertainties in the CSD indicators (or other higher-order statistics). The only uncertainty provided for EN4.2.2 is the uncertainty associated with the analysis method, and that estimate has issues that limit its usefulness for our analysis; for example, in some areas more observations actually increase the analysis uncertainty (see[29]). Thus, in this section, we make use only of the uncertainties provided with the HadCRUT5, HadSST4 and ERSSTv5 datasets; see Methods for a detailed discussion of the analysis and processing procedures of these datasets, as well as of the provided uncertainties.

We first use all three datasets to calculate the AMOC fingerprint proposed by Caesar et al. 2018[12] (Fig. 1). The advantage to using all three datasets is that they represent three different ways of dealing with missing observations. HadSST4 simply has no data where there are no observations. In the HadCRUT5 infilled dataset those data points are filled in, with the exception of a few remaining gaps, and in the ERSSTv5 dataset all data points are filled in. Thus, HadSST4 only has continuous coverage of the SPG from 1873, after which the number of gridcells with data in each month gradually increases (Fig. 2c), whilst HadCRUT5 has about the same number of grid cells in the SPG over the whole period (Fig. 2e), and ERSSTv5 has full coverage throughout (not shown). Thus, in HadSST4 the AMOC index could be biased towards a sub-region of the SPG with more observations in certain time periods. The increasing number of averaged full grid cells can affect the variance due to the increased signal-to-noise ratios with higher data density. On the other hand, infilling the missing grid cells in HadCRUT5 and ERSSTv5 comes with the uncertainties of the respective infilling methods (see Methods).

For HadSST4, we create an ensemble that captures the uncertainty at each grid cell by sampling the provided uncertainties (see Methods) around each bias ensemble member, creating 200x100 samples. For HadCRUT5 and ERSSTv5, we simply make use of the provided 200-member and 1000-member uncertainty ensembles, respectively. In HadCRUT5 this ensemble already accounts for all aspects of uncertainty, including the interpolation uncertainty. In ERSSTv5 the ensemble only accounts for the parametric uncertainty, but since many parameters are associated with the reconstruction this ensemble includes the data processing effects that can influence the detection of CSD (see Methods). In all three cases, we calculate the AMOC index following Caesar et al.[12] for each sample or member and then calculate its $\lambda$ (Fig. 1c, d). Because of the infilling of missing values, the HadCRUT5 and ERSSTv5 based indices and corresponding $\lambda$ estimates have a larger uncertainty range than for HadSST4, especially in the early years. This is also reflected in the distribution of trends: the fitted Gaussian of $\lambda$ trends of the HadSST4 samples is much narrower ($\mu = 0.0059, \sigma = 0.0003$) than the ones for HadCRUT5 and ERSSTv5 ($\mu = 0.0046, \sigma = 0.0013$ and $\mu = 0.0049, \sigma = 0.0007$, respectively)(Fig. 1e,f). We show the median, mean and operational timeseries for HadSST4, HadCRUT5 and ERSSTv5, respectively, as those are the time series a user would obtain when downloading the data from the Met Office and NOAA websites (www.metoffice.gov.uk/hadobs/hadcrut5/, www.metoffice.gov.uk/hadobs/hadsst4/,

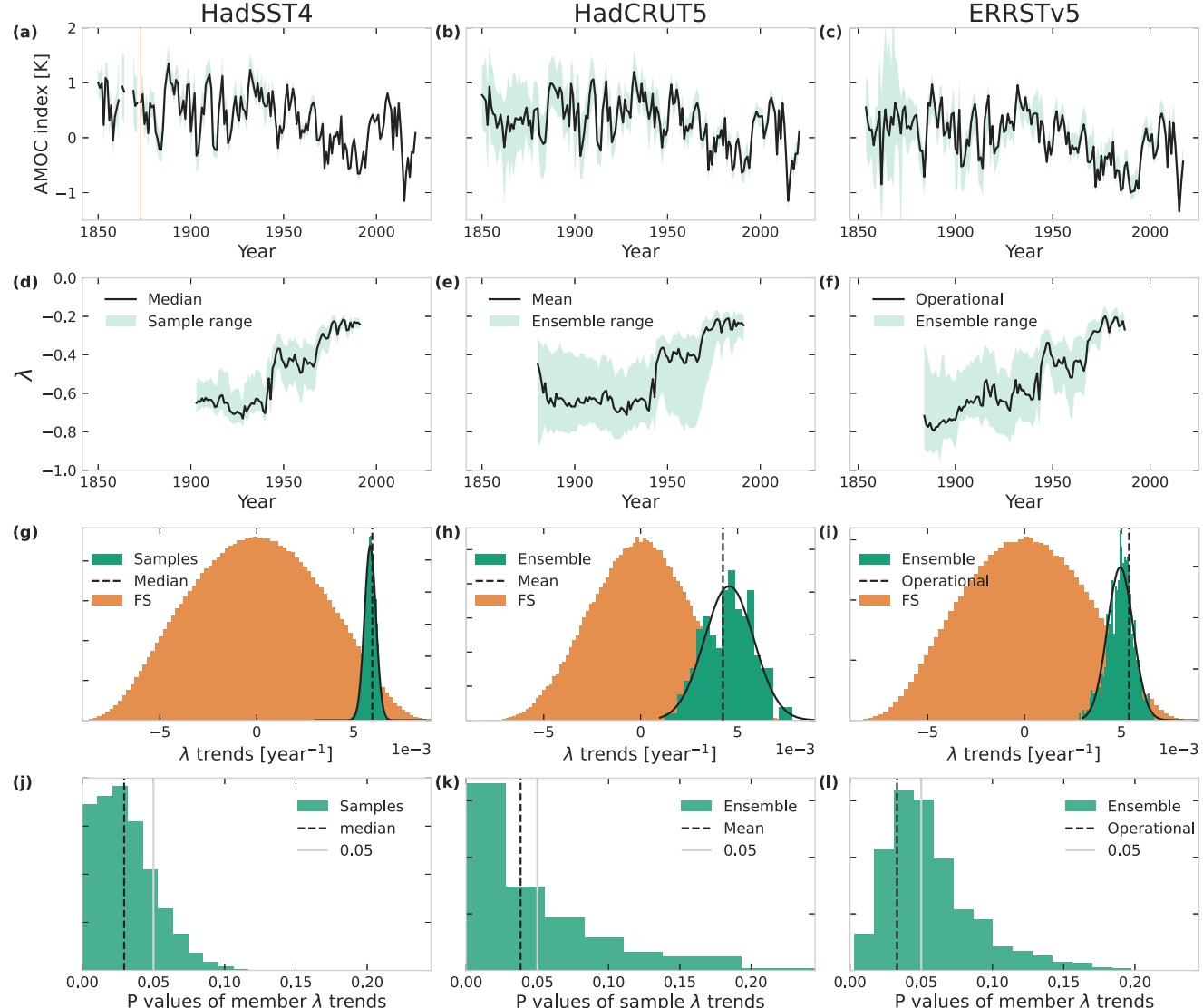

**Fig. 1 | Uncertainties of the restoring rate $\lambda$ for sea-surface-temperature-based fingerprints.** Significance of trend in the restoring rate $\lambda$ of subpolar gyre Atlantic Meridional Overturning Circulation (AMOC) index in HadSST4 (**a**, **d**, **g**, **j**), HadCRUT5 (**b**, **e**, **h**, **k**) and ERRSTv5 (**c**, **f**, **i**, **l**). **a**, **b**, **c** Median (mean) AMOC index (black) and min-max range (turquoise) of 20000 samples (200 and 1000 ensemble members) for HadSST4 (HadCRUT5 and ERSST). We use the mean instead of the median for HadCRUT5 as that is the default product a user would download when not investigating uncertainties, and for ERRSTv5 we use the operational data product (see main text). **d**, **e**, **f** Same as (**a**, **b**, **c**) but for the $\lambda$ of the AMOC indices, computed using a window size of 60 years. **g** Distribution (orange) of linear trends of $\lambda$ computed from 100 Fourier surrogates for each AMOC index sample from HadSST; distribution (turquoise) of linear trends of $\lambda$ of samples from HadSST4 with a fitted gausian distribution (solid black). The linear trend of the median HadSST4 index is shown in dashed black. **f**, **i** Same as e but for HadCRUT5 and ERRSTv5, using 1000 fourier surrogates (FS) of the ensemble member AMOC indices, and the linear trend of the mean and operational data. **j** p-value of linear trend of $\lambda$ of each sample of the HadSST4 AMOC index with respect to its own 100 Fourier surrogates. The p-values of the median with respect to 10000 Fourier surrogates is shown as a dashed black line, and the 0.05 significance value is shown as a solid grey line. **k**, **l** Same as g but for the AMOC index of the HadCRUT5 (ERSST) ensemble members, with the p-value of the mean (operational) AMOC index as a dashed black line.

---

https://psl.noaa.gov/data/gridded/data.noaa.ersst.v5.html).The operational product in ERSSTv5 is produced using the operational values of the parameters, as opposed to their perturbed versions in the parametric ensemble (see Methods).

However, the magnitude of the trend is of lesser interest to us than whether or not it is statistically significant. To test for significance, we calculate 1000 Fourier surrogates (see Methods) from each AMOC time series, and use the obtained linear trends from the $\lambda$ time series for each sample or member individually to calculate a p-value (Fig. 1g, h). 85.56% of HadSST4, 65% of HadCRUT5 and 50.8% of ERSSTv5 p-values are below 0.05, and 99.7%, 84.5% and 90.1% are below 0.1, showing that even considering the dataset uncertainties, the increase in $\lambda$ of the SPG-based AMOC index is significant.

For the HadCRUT5 and ERSSTv5 datasets, we can take the analysis a step further - most regions of the Atlantic are either completely infilled or have enough years of data (Fig. 2e) to compute $\lambda$ for each grid cell, and repeat this for all ensemble members. The resulting map shows a significant positive linear trend of $\lambda$ in the North Atlantic (Fig. 3a, i), similar to that seen in B21 for the HadISST dataset[5]. The trends in the ensemble members vary, but their mean is very similar to the trend of the ensemble mean (Fig. 3b, j, c–h, k–p). We also fit a Gaussian curve to the ensemble distribution at each gridpoint and show the regions where most of the uncertainty ensemble $\lambda$ trends are positive ($\mu - 2\sigma > 0$, -95%). These regions are along the northern subtropical gyre, the North Atlantic Current and in the Greenland, Iceland and Norwegian Seas in

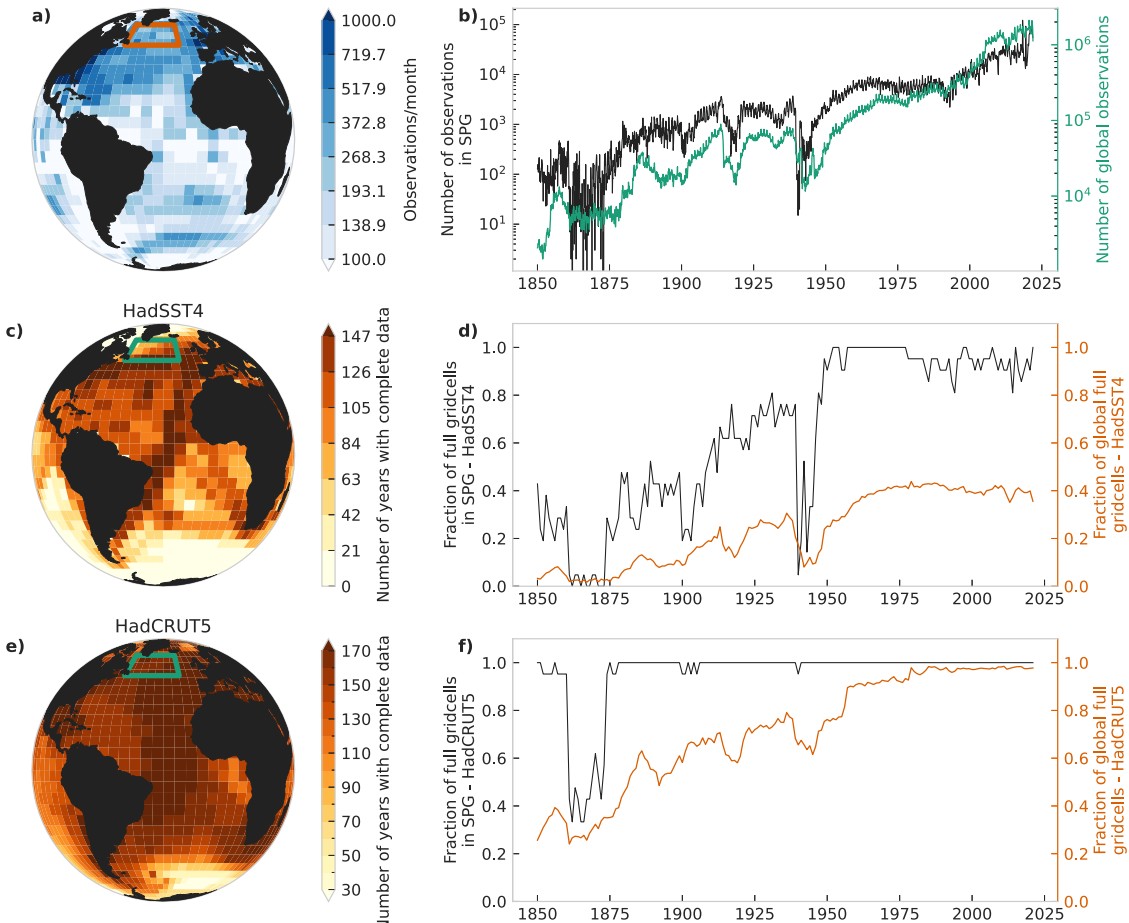

**Fig. 2 | Non-stationarity of sea-surface temperature (SST) observational data coverage. a** Mean number of SST observations per month in HadSST4 and Had-CRUT5 for the time span 1850–2021 in each grid cell. **b** Number of observations per month in HadSST4 and HadCRUT5 in the subpolar gyre (SPG, black) and globally (turquoise). Raw observations are the same for both HadSST4 and HadCRUT.

**c** Number of years with complete annual mean data (see Methods) for HadSST. **d** Fraction of full grid cells (grid cells with data in each month), after taking the annual mean in the SPG (black) and globally (orange) in HadSST. **e** Same as (**c**) but for HadCRUT5. **f** Same as (**d**) but for HadCRUT. The SPG area is shown as a square on the maps.

HadCRUT5 and ERSSTv5, with the addition of the sub-polar gyre (Fig. 3a, i).

Supplementary figures 1, 2, 3 and 4 show the same calculations for the variance and autocorrelation. The autocorrelation behaves similarly to $\lambda$, although with lower p-values. The variance has no overall trend, due to a marked decrease from 1850 to the 1950s in all three SST datasets, which is likely caused by the increasing number of observations for the reasons explained above.

### Global significance estimation with surrogates

In addition to an uncertainty estimation, it is also important to calculate the statistical significance of our higher-order statistic of interest. When we want to test a statistic of a time series $x_t$, for example the linear trend of its lambda time series, denoted by $s_\lambda(x_t)$, this is generally done by generating surrogate time series of $x_t$, $\bar{x}_t^i$[33]. The values of the statistic for many surrogate time series $s_\lambda(\bar{x}_t^i)$ can then be used as a distribution to estimate the significance of the actual $s_\lambda(x_t)$. Our null model determines the properties that the surrogate time series need to have. When calculating CSD indicators, our null model should be a time series that has the same autocorrelation structure and variance as $x_t$, but is otherwise random. Such a time series can be produced either by measuring a finite number of autocorrelation properties of $x_t$ and generating an according time series, or by the method of Fourier surrogates, where the Fourier phases are randomly shuffled (see Methods). These surrogates can be modified to include the effects that

interpolation methods or lack of data have on the time series (e.g. removing data points that are missing in the original time series).

It is important to note that in the case of calculating CSD indicators, our conservative null model is an SST or salinity time series with given properties, not a $\lambda$ time series. In this regard, the surrogate analysis of this work is more conservative than that in B21, who considered surrogates of the $\lambda$ time series. This is because the autocorrelation structure of $\lambda_t$ has a non-trivial dependence on the autocorrelation of $x_t$. By generating surrogates of $\lambda_t$ as in B21 one ignores the wider range of autocorrelations that $\lambda_t$ can have given the autocorrelation of $x_t$. This can result in a narrower distribution of surrogate $\lambda$ trends, and the significance of $s_\lambda(x_t)$ is thus overestimated. It is therefore important to generate time series from $x_t$ and not $\lambda_t$, even if the former is computationally more costly. Note that this also implies that Fourier surrogates are not generally suited to test trend significance in arbitrary time-correlated time series. They should only be used in situations where the trend of a sliding-window higher-order statistic is estimated from a given time series, and one has access to that time series to compute surrogates from.

In addition to the SST datasets with uncertainty estimates used in the previous section, we also apply the surrogate analysis to HadISST1 for consistency with B21. The HadISST and ERSSTv5 datasets are globally interpolated, and thus the time series at each gridpoint is complete, and we can use Fourier surrogates to calculate their regions of significance (Fig. 4a). Due to the holes in the HadCRUT5 dataset we

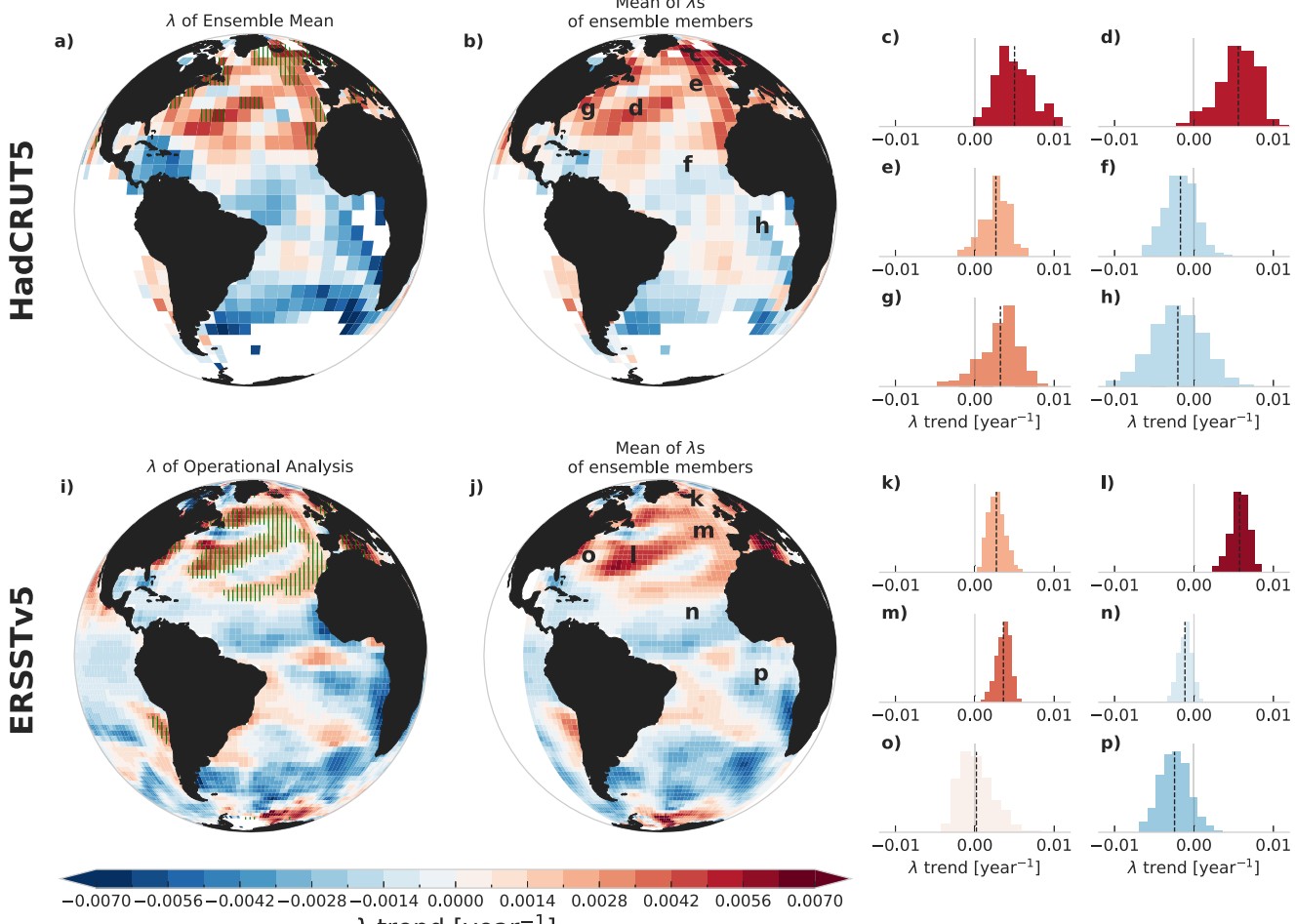

**Fig. 3 | Spatial fields and distributions of trends of the restoring rate λ for sea-surface temperatures. a** Linear trends of restoring rate λ time series computed from the ensemble mean in the HadCRUT5 (**a–h**) and ERSSTv5 (**i–p**) datasets. b. Mean of the linear trends computed individually from each λ time series of the 200 ensemble members. **c–h** Distributions of 200 trends for locations marked with corresponding letters in (**b**), with a vertical black dashed line showing the ensemble mean, and light solid grey lines at 0. The green vertical hatches in (**a**) indicate the regions where μ − 2σ > 0 for the Gaussian fitted to the ensemble distribution, i.e. where ~95% of the uncertainty ensemble trends are positive. i-p. same as a-h for the 1000 ensemble members of the ERSSTv5 dataset. Note that the shaded regions in a and i do not represent regions of statistically significant trends, but rather the regions that are increasing in most ensemble members.

cannot use Fourier surrogates (see Methods) and thus use AR(2) surrogates instead to calculate the significant regions (Fig. 4b).

For all three datasets, the regions of increasing λ extends over the whole North Atlantic with similar patterns, but only smaller regions show statistically significant change at the 0.05 confidence level. For HadISST, significant increase only occurs along the North Atlantic Current, the density-driven part of the AMOC, whilst for HadCRUT5 the significant increase occurs at the northwestern edge of the sub-tropical gyre, along the North Atlantic Current and in the eastern sub-polar North Atlantic, including the Greenland, Iceland and Norwegian seas. For ERSSTv5 the regions of significance are much smaller, with only one major region around the northern subtropical gyre.

As opposed to the SST datasets, the infilling method in EN4.2.2 uses information from previous times, as well as a climatology. This has the potential to cause false indications of CSD (see Methods for a full discussion). Thus when generating surrogates for a time series from the EN4.2.2 salinity dataset, we must consider the effect the lack of data and analysis method has on the earlier years. The full analysis process is too complex to reproduce when generating the surrogates. However, we can reproduce the specific effect that the analysis procedure has on the calculated CSD indicators. To do this, we use the observational weights provided with the dataset (see e.g.

Supplementary Fig. 5h–m). These observational weights were produced by setting all observational values to one and the climatology to 0 and rerunning the analysis that produced the infilled dataset[29]. The resulting weight $w$ represents the amount of information in the given analysis value that comes from observations. It should be noted that this observational information comes from the whole globe, and so $w$ can be high even if there is no observation at the specific gridpoint at that time. However, a low $w$ value is still a good indicator of a datapoint where the persistence based forecast dominates.

For each time series, we use the autocorrelation properties of its last 40 years to generate AR(2) surrogates. Then, for each month that has an observational weight below some limit $w_0$, we replace the surrogate value with the persistence-based forecast (Eq (1)). This creates the same spikes in the surrogates that are present in the analysis data, and thus modifies the autocorrelation structure in a similar way (Supplementary Fig. 7).

Using the unmodified surrogates, there are regions of significantly positive linear trends of λ for the EN4.2.2 salinity data at the northern edge of the sub-tropical gyre and along the North Atlantic Current (Supplementary Fig. 8e). The modification of the surrogates to account for the non-stationary data coverage results in spurious increasing and decreasing λ trends (Supplementary Fig. 8b–d), with

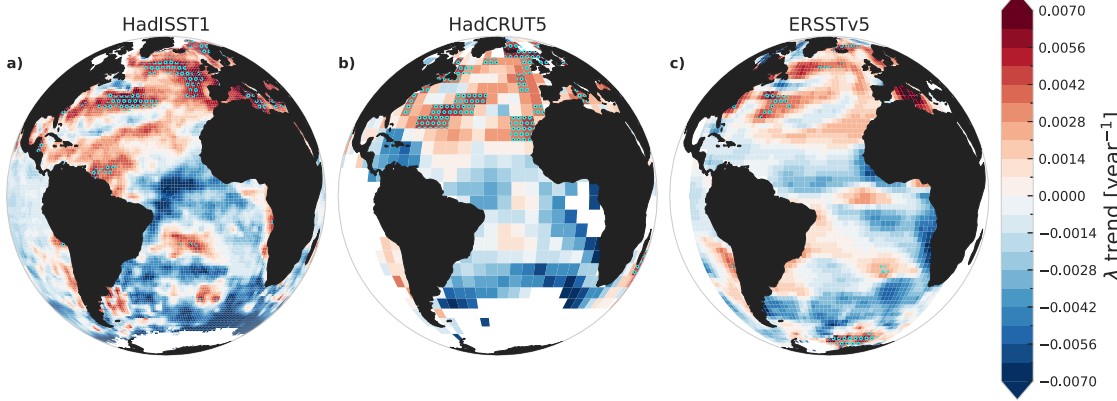

**Fig. 4 | Statistical significance of trends of the restoring rate λ calculated from the sea-surface temperature uncertainty ensembles. a** Linear trends of restoring rate λ time series for the HadISST1 data. **b** Same as (**a**) but for the HadCRUT5 mean dataset. **c** Same as (**a**, **b**) but for the ERSSTv5 operational dataset. Light blue stippling shows the regions where the positive trends are significant at 95th percentile, calculated from 1000 Fourier surrogates for each cell for HadISST (**a**) and ERSSTv5 (**c**) and 1000 AR(2) surrogates for each cell for HadCRUT5 (**b**). See Methods for more details.

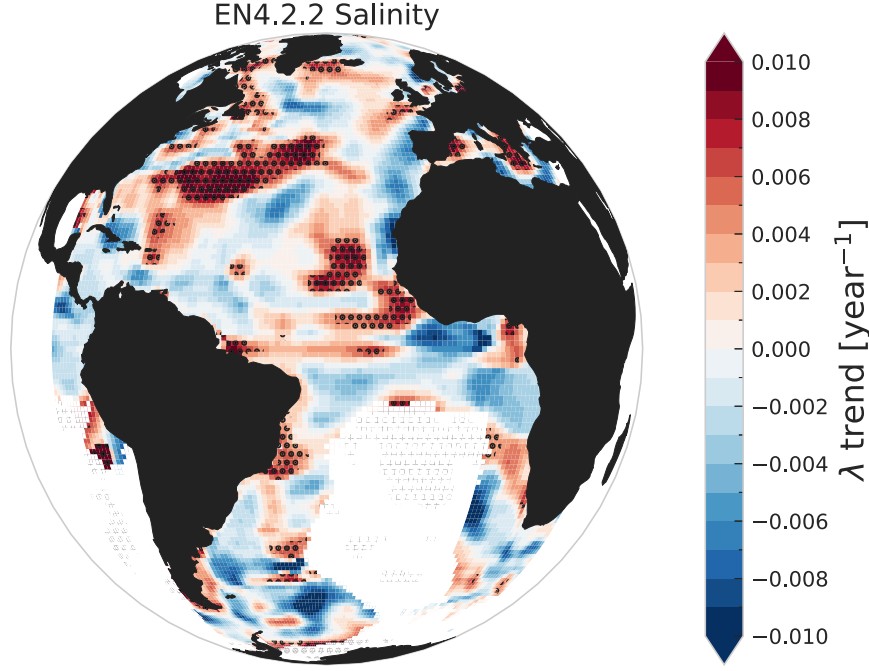

**Fig. 5 | Statistical significance of trends of the restoring rate λ calculated from the EN4.2.2 salinity dataset.** A Linear trend of restoring rate λ time series for the average top 300m salinity in the EN4.2.2 dataset. Black stippling shows the regions of 95th percentile significance in the positive trends calculated using 1000 AR(2) surrogates modified following the EN4.2.2 analysis method with an observational weight bound of 0.5. These significance regions are indistinguishable from those calculated from the unmodified surrogates (see Supplementary Fig. 8). Only regions between 90°W and 30°E with 60+ years of observational weight above 0.05 are shown. See Supplementary Fig. 6 and Methods for details.

slightly positive trends in the North Atlantic and negative ones near the equator, as would be expected from the autocorrelation values of those regions. However, the magnitude of these false trends is much smaller than the increases seen in the analysis data in some regions, and thus the regions of significance in the Atlantic Ocean remain basically unchanged when we use the modified surrogates (Fig. 5). This is true regardless of the limit value chosen for $w_0$ (Supplementary Fig. 8f–h).

The global surrogate analyses for variance and autocorrelation can be found in Supplementary Figs. 9, 10, 11 and 12. The results for the autocorrelation are again similar to the results for λ estimated under the assumption of non-stationary correlated noise. But although the regions of positive trends in the North Atlantic are spatially coherent with those of λ, the regions with significant ($p < 0.05$) increase are instead at the northern Gulf Stream and its extension into the Atlantic Ocean (Supplementary Figs. 13, 14). The variance of each dataset has a different result for the trends and their areas of significance (Supplementary Fig. 10, 12). The possible causes of different results for the SST datasets are discussed in the Methods. In contrast to the autocorrelation and λ, the surrogates calculation for the variance in EN4.2.2 (Supplementary Fig. 12) is a clear example of the utility of modifying the AR2 surrogates to match the analysis method. Without modification the whole of the Atlantic seems to have a significant increase in variance. But once the effect of the analysis method is incorporated, no significant regions remain, and we recognise the increase in variance as spurious, caused by the analysis procedure of EN4.

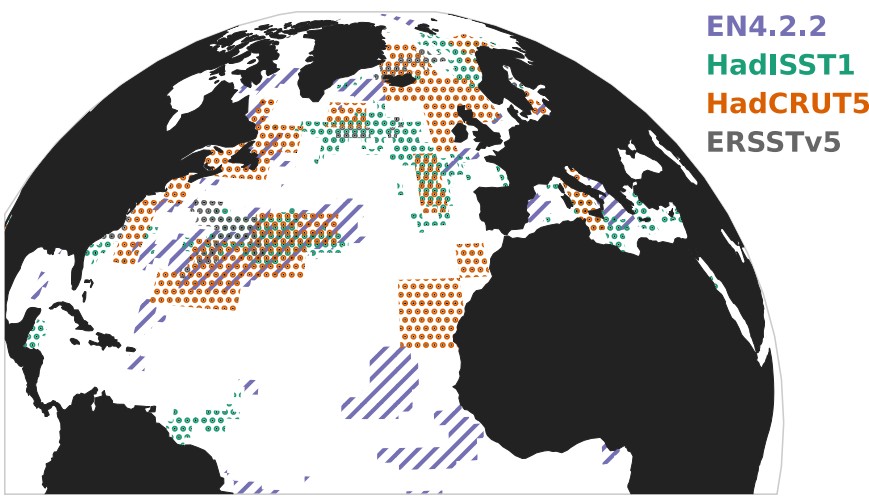

**Fig. 6 | Regions in the North Atlantic with a statistically significant increase in the restoring rate λ for sea-surface temperatures (SSTs) and salinity concentrations.** For the EN4.2.2 salinity (purple), HadISST SSTs (turquoise), HadCRUT5 SSTs (orange) or ERSSTv5 SSTs (grey). Significance is calculated from 1000 surrogates for each grid cell (see text and Methods).

## Discussion

We have addressed a common problem that arises when CSD indicators are computed from pre-processed observational data, namely, that the observational datasets have inherent and potentially non-stationary uncertainties and biases that could influence the analysis[8]. As well as complementing the analysis of B21, this work can thus be used as a basis for observational uncertainty analysis of other higher-order statistics.

Using the uncertainties provided with the datasets, we estimate an uncertainty range of the linear trend of λ in the SST-based AMOC index (Fig. 1) of 0.0059 ± 0.0003 for the HadSST4, 0.0046 ± 0.0013 for the HadCRUT5 dataset and 0.0049 ± 0.0007 for the ERSSTv5 dataset, respectively. We have also updated the surrogate significance analysis of B21 for the global SST and salinity data (HadCRUT5, HadISST1, ERSSTv5 and EN4.2.2), and find that our more conservative significance test reduces the area of significant CSD indicators compared to that in B21. However, the four datasets show significant λ increases in the area of the northern subtropical gyre, the North Atlantic Current and in the Greenland, Iceland and Norwegian Seas (Fig. 6).

Whilst our work demonstrates that the indication of CSD in the chosen AMOC fingerprints is not due to the reported inherent properties of the observational datasets, it is beyond the scope of this work to investigate whether or not this is a clear indication that the AMOC is destabilizing. In fact, the choice of SPG SSTs as an AMOC fingerprint is still debated[34,35]. This is because the extent that the AMOC controls the temperatures in the so-called SPG warming hole is still uncertain[36-40]. Although the SPG-based SST fingerprint has been shown to have a lagged correlation with the AMOC streamfunction strength[35,41], the strength of the correlation seems to be forcing and model dependent[34,41]. These caveats, together with studies showing that this fingerprint is noisier than others[35], mainly question the fingerprint's ability to detect a change in the mean state of the AMOC. However, these criticisms should be less relevant for a CSD analysis, as that is only focused on the statistics of the perturbations from the mean state, not on the mean state itself or on the source of the perturbations. However, there are yet to be any studies showing that signs of destabilization along the North Atlantic Current or in the SPG are in fact signs that the large-scale AMOC is destabilizing. In fact, the North Atlantic SPG has recently been identified as a separate tipping element[42,43], and thus some of the identified CSD indicators may be caused by only this subset of the Atlantic Ocean approaching a transition.

These uncertainties in the SPG fingerprint are the reason that we also look for CSD in every gridcell in the Atlantic (see Figs. 3, 4, 6), allowing us to investigate their spatial patterns in detail. However, there is still the possibility that the detected increase in λ is not caused by an AMOC destabilization. Processes that act on shorter time-scales, such as air-sea heat fluxes, act as the perturbing noise and changes in their characteristics are accounted for in our calculation of λ (in contrast to the calculation of the classical CSD indicators, i.e. the variance and AC1, see Methods). But there could be other long-term changes in internal ocean processes that would cause an increase in the restoring rate. For example, a gradual increase in mixed layer depth would increase the effective heat capacity and possibly with it the response time. However, we consider it implausible that this effect explains the long-term trend in λ, as the ocean mixed layer is not observed to be deepening overall and global warming is in fact expected to increase the stratification of the upper ocean[32]. In addition, such damping of fluctuations would not only increase λ but also decrease the variance, and we find that in the subpolar North Atlantic the variance has actually been increasing in the most recent decades, although this result is more sensitive to the dataset properties than the increase in λ (Supplementary Fig. 2).

In addition, the fact that the areas of significant CSD indicators on the map are much smaller in our analysis than in B21 strengthen the case that these trends are caused by an AMOC destabilization. In B21, the whole North Atlantic was marked as significant, as well as large regions in the South Atlantic. In this work, the significance is reduced to smaller regions in the North Atlantic that are more typically associated with the path of the warm branch of the AMOC: the northern subtropical gyre, the North Atlantic Current, the Irminger Sea and the Greenland, Iceland and Norwegian Seas. These CSD indicators could be a sign of AMOC destabilization, as the SST and salinity in these regions would be sensitive to the strength of the AMOC[15]. The Irminger and Iceland basins in particular have been recently identified as the centres for subpolar AMOC variability, as opposed to the subpolar gyre[44,45]. This could be the reason that λ in the regions used to define the SST index is increasing, but not significantly, as opposed to the significant increase in the Greenland, Iceland and Norwegian Seas (Fig. 6). This may seem to contradict the significant increase found in the AMOC fingerprint obtained by averaging SSTs over the SPG (Fig. 1), but in fact when averaging the SST over an extended region one gets rid of local processes and improves the observational accuracy, allowing for the overall trend in λ to become significant. This is the strength of combining both averaged indices and geographic

cell-by-cell analyses. Finally, note that the definition of significance on the map in this work (Figs. 4, 6) is highly conservative. Significance is calculated on a point-by-point basis and not globally, and we do not take the spatial coherence of positive trends in the North Atlantic into account in our significance testing.

Together with the computational expense, the uncertainty estimates provided with the observational datasets are the most important factors for a reliable uncertainty estimation of CSD. Although running the full analysis algorithm that is used to create the observational datasets is beyond the scope of this work, we are able to make modifications to the surrogates that influence the statistical properties of the data in a similar way to the full analysis. However, we cannot estimate the complete uncertainty on the CSD indicators of the salinity dataset EN4.2.2 because the analysis but not the observational uncertainties are provided. Even so, our results show that the influence of observational analysis methods on higher orders statistics should be taken into account alongside the effect on the long-term mean properties.

In summary, for CSD indicators computed from observation-based SST and salinity data we have presented a comprehensive uncertainty estimation and propagation together with significance testing. Such an analysis is a prerequisite to robustly assessing the destabilization of a system from observational data. We find that data processing methods can lead to false detection of CSD (see also[8]). However, we demonstrate that such obstacles can be overcome by incorporating the data processing effects into uncertainty estimates and significance testing.

## Methods
### HadISST
The HadISST dataset is based on the Met Office Marine Data Bank as well as the Comprehensive Ocean-Atmosphere Data Set (COADS)[46], and has been bias adjusted and then temporally and spatially homogenized using reduced-space optimal interpolation (RSOI). RSOI uses a set of global empirical orthogonal functions (EOFs), and includes regularizing terms when fitting the EOFs to the data. This is done to avoid spurious large amplitudes in data-scarce regions and times, but means that the fit tends to the zero anomaly where there is no information. Although non-interpolated in-situ data is subsequently added to the RSOI reconstruction, this only improves the variance where there is enough data, and thus in data-scarce times and regions the variability is damped by RSOI. Together with other steps of the preprocessing, this causes the variance in HadISST1 to artificially increase with time. This infilling thus means that the dataset is not optimal for statistical analyses of climate variability. In addition, HadISST does not have an uncertainty estimation of either the bias adjustement, analysis method, measurement or sampling uncertainty. This makes it difficult to estimate possible effects on the CSD analysis[27].

### HadSST4 and HadCRUT5
Given the lacking uncertainty information for HadISST we focus primarily on three similar SST datasets, namely HadSST4[30] and the SST part of HadCRUT5[31], as well as ERSSTv5 (see next section below). HadSST4 is based on observations from the International Comprehensive Ocean-Atmosphere Data Set (ICOADS[46],) gridded to a 5° by 5° grid. Various bias adjustments are applied to the data to account for the changes in historical SST measurement techniques. The HadSST4 dataset has a 200 member ensemble which explores variations of the bias scheme parameters, and in addition comes with measurement and sampling uncertainties. The measurement uncertainty is associated with the measurement error, and the sampling uncertainty estimates the uncertainty arising from the under-sampling of the data, and scales as the inverse of the number of observations. This uncertainty analysis is ideal for estimating uncertainties of CSD indicators.

However, as the HadSST4 dataset is non-infilled, it has large gaps where there is no data. This makes a grid cell by grid cell CSD analysis impossible, and even causes difficulties when averaging data over larger regions such as the subpolar gyre. In this work, we therefore complement our analysis with SSTs from the HadCRUT5 dataset. The SST data in HadCRUT5 is based on the HadSST4 dataset, but provides an additional, more globally complete analysis dataset. The gaps in the data are filled using a Gaussian-process-based statistical method. In regions where the local observations offer an insufficient constraint this infilled data is removed, so the final dataset still has some gaps (see Fig. 2). The dataset is comprised of 200 ensemble members, which sample the reconstruction error for the Gaussian process in addition to the bias and observational uncertainty in the data.

The number of individual SST observations has increased approximately exponentially over the last 150 years (Fig. 2). This increase in observations affects the statistical properties of the data. Primarily, an increase in observations might cause a decrease in the variance, as the data in later times is an average of more values and the variance of the mean scales as $\sim \frac{\sigma^2}{n}$, where $\sigma^2$ is the variance of the individual observations and $n$ is the number of observations[30]. The effect this would have on the autocorrelation is more difficult to determine. The more accurate values in later times could cause a higher autocorrelation due to the improved signal-to-noise ratio of the data (see[8]), but the larger range of measurement instruments used in later times could also reduce any false contributions to the autocorrelation that are related to the instruments. In all these cases, a large part of the effect would be included in the uncertainties provided with the datasets, as they account for both sampling and measurement uncertainties.

### ERSSTv5
The ERSSTv5 dataset[47] is based on the same set of observations as HadSST4 and HadCRUT5 (ICOADS version 3.0[46]). After quality control and bias correction, an interpolation method using empirical ortho-gonal teleconnections (EOTs) is used to make the dataset globally complete. This is done by first seperating the data into low and high frequency components using a moving filter of 26° by 26° and then a median filter of 15 years. The high frequency component is then decomposed with 140 EOTs, which were trained on the 1982-2011 period of the National Centers for Environmental Prediction (NCEP) Weekly Optimum Interpolation SST (WOISST) dataset[48]. EOTs are similar to empirical orthogonal functions, but are restricted spatially. The high frequency component of the SST data is then reconstructed using only those EOTs that are not undersampled by the data. In the low frequency component the missing values are filled in with the running average of the 26° × 26° region when the ratio of observational coverage within the region reaches a minimum criterion. The EOT reconstruction inevitably leads to a loss of information, and to greater smoothing when data is scarce.

There are two uncertainty products provided for ERSSTv5: an ensemble that samples the parametric uncertainty, and a climatological reconstruction uncertainty. The former is calculated by sampling 1000 combinations of the 28 most important parameters of the data processing method, and repeating the whole data processing procedure with the different parameter combinations. The reconstruction uncertainty is calculated by applying the reconstruction method to pseudo-observations and calculating the difference of the resulting dataset from the original. The reconstruction uncertainty is only a climatology, and is thus constant throughout the dataset time period. This means that it is not informative for our purposes, since we are interested in data processing effects that change over time. In fact, the time-varying uncertainty of the reconstruction is included in the parametric uncertainty ensemble, as many of the parameters varied are associated with the reconstruction method. We thus opt to use only the parametric uncertainty ensemble in this study.

## EN4.2.2

The statistical properties of the EN4.2.2 dataset are affected by the data analysis method in a much clearer way than the other datasets. EN4.2.2 includes both global quality-controlled ocean temperature and salinity profiles and monthly objective analyses[29]. The profiles are direct observations from various sources, such as the World Ocean Database[49]. They are used to obtain the globally complete analysis by calculating an optimal fit to the good profiles and profile levels in each month, given a background (prior constraint). Good profiles and levels are those which do not fail any quality-control check. The resulting optimal interpolation equations are solved using a numerical scheme. The background used for this calculation is a damped persistence-based forecast:

$$\mathbf{x}_i^f = \mathbf{x}_i^c + \alpha(\mathbf{x}_{i-1}^a - \mathbf{x}_{i-1}^c), \tag{1}$$

where $\mathbf{x}_i^f$ denotes the damped persistence forecast for month $i$, $\mathbf{x}_i^c$ is the climatological mean for that month, and $\alpha = 0.9$. As we are concerned here with the statistical properties of the data, the influence of using such a persistence-based forecast as the background needs to be addressed. If there are no observations for long periods of time, the analysis will relax to the climatology for that given location. If we then have a single observation, this causes a spike in the data, which relaxes back to the climatology with a monthly lag-1 autocorrelation of 0.9. In most of the Atlantic Ocean there are very few observations before the 1950s (see Supplementary Fig. 5), and so the monthly autocorrelation is artificially forced to about 0.9 at the start of the time series, which also affects the yearly autocorrelation. Depending on the true autocorrelation function of the underlying time series and depending on how much of the forecast is used, this analysis method causes a biased estimate. In particular, it could cause a false indication of CSD if the true monthly autocorrelation for the more recent decades is systematically above 0.9.

## Gaps in the data

Both HadSST4 and HadCRUT5 have gaps in their time series. When averaging this data for different regions, we take a conservative approach: We first spatially average the monthly resolution data and then take the yearly average of the resulting time series. When taking the yearly average we only consider a year if there is data for all 12 months, otherwise it is set to NaN. However because we make the monthly-resolution spatial average first, this approach does not ensure that each grid cell in the region has data for each month of the year. Thus the value for a given year might have more grid cells contributing to one month than another, which could affect variability. For HadCRUT5 we also calculate CSD indicators at each grid cell where there are less than 30 missing years out of 172.

## Critical slowing down indicators

The restoring rate $\lambda$, the variance and the autocorrelation are calculated in the same manner as in B21. Each time series is first nonlinearly detrended using a running mean with a 50-year window. The edges are not removed, so the detrending method is less certain at the first and last 25 years of the time series. The CSD indicators are then calculated in 60-year running windows. The variance and autocorrelation are calculated in the standard way. The restoring rate is calculated by regressing $\Delta x_i$ against $x_i$ using the GLSAR function from the python module statsmodel. Note that as in B21 the $\lambda$ plotted in this study is the numerical result of the regression of $\Delta x_i$ against $x_i$, and so is related to the analytical $\lambda'$ defined in the text by $\lambda = e^{\lambda'} - 1$ (when the timestep $\Delta t = 1$). As the magnitude of $\lambda$ is immaterial in this study and we are only concerned with its increase or decrease, both definitions behave similarly and are thus interchangeable for our purposes.

## AMOC indices

The SST-based AMOC index is calculated as the mean SST in the subpolar gyre minus the mean global SST, following Caesar et al.[12]. The subpolar gyre in this work is taken as the area between 41° and 60° N and 20° and 55° E (following[50] for ease of calculation). This is a slightly different area than that used by B21, but makes little difference at the low 5° resolution of HadSST4 and HadCRUT. We also take the full year instead of the winter months, as the latter method causes no substantial difference for the change in statistical properties. The index is thus the average annual SST in the defined rectangular SPG area minus the global mean annual SST. Note that we only calculate the index for HadCRUT5, HadSST4 and ERSSTv5, and since these datasets have their own methods of dealing with sea ice cells we do not impose any sea ice masking.

All salinity time series and global plots in this work are for the thickness-weighted mean of the upper 19 ocean layers, corresponding to the average salinity in the top 300 m of the salinity profiles. The observational weights are similarly averaged in the top 19 layers. The uncertainty is calculated by simple uncertainty propagation: $\Delta s = \sqrt{\sum \Delta s_i^2}$, where $\{\Delta s_i\}$ are the uncertainties of each individual level. This incorrectly assumes the layer uncertainties are independent, but is acceptable here as the uncertainty is not used for any quantitative analysis, but only to display an uncertainty range in Supplementary Fig. 5.

## Surrogates

Surrogates are created from the detrended SST and salinity time series. Fourier surrogates are calculated by taking the discrete Fourier transform of the time series, multiplying by random phases and then taking the inverse Fourier transform. By the Wiener-Khinchin theorem, the variance and autocorrelation function of wide-sense-stationary random processes are specified by the squared amplitudes of the (discrete) Fourier transform. Thus the Fourier surrogates preserve the variance and autocorrelation function of the original time series.

However, in this work we know the time series we are dealing with have been modified by the analysis process and lack of observations. When the analysis method modifies the autocorrelation, as in the case of the salinity data, the Fourier surrogates of the full time series are not a correct null hypothesis for CSD analysis, because the autocorrelation function will include information from the earlier, modified times. In addition, Fourier surrogates can only be calculated for time series with no missing values, and thus cannot be used for the HadCRUT5 global analysis. Thus, in these cases, we do not use Fourier surrogates.

## AR(2) surrogates

For the cases where Fourier surrogates cannot be computed, we choose to use AR(2) surrogates on a monthly resolution:

$$x_t = a_1 x_{t-1} + a_2 x_{t-2} + \epsilon_t, \tag{2}$$

where the time series value at time $t$, $x_t$, is determined by the value at times $t-1$, $t-2$ with autocorrelation coefficients $a_1$, $a_2$, and $\epsilon_t$ is white noise. Even though the monthly time series in the datasets are close to being AR(1) processes, the higher coefficient is needed to get the correct lag-one autocorrelation coefficient for the yearly averaged time series. Since the yearly average is taken before calculating CSD indicators, using an AR(2) process instead of AR(3)-AR(12) does not make a big difference as long as the annual lag-one autocorrelation is correct. If the estimate of monthly lag-one autocorrelation is $A_m$, the coefficients are related by:

$$a_1 = A_m(1 - a_2) \tag{3}$$

We get the values of $a_1, a_2$ for each time series by calculating the true lag-one autocorrelation estimate for the last 40 years of the annual and monthly time series, $A_y, A_m$, and then calculating the $a_2$ value that minimizes the difference of the estimated $A'_y, A'_m$ of the AR(2) time series from the true values.

## Modification for salinity surrogates

It is only possible to produce perfect surrogates for the salinity analysis data by repeating the complete analysis used to create the dataset. This is not feasible for this study. We can, however, replicate the effect of the analysis method that would potentially cause spurious CSD detection, namely the relaxation back to the climatology that occurs when there is a lack of observational data. For this we utilize the observational weights provided as part of the EN4.2.2 dataset. We start with an AR(2)-based surrogate dataset of the global monthly data averaged over the levels in the top 300 m. For each grid cell, we take the months that have an observational weight that is below some limit $w_0$ and replace the surrogate value with the persistence-based forecast (see Eq. (1)). As we did not have access to the climatology used in the EN4.2.2 analysis, we took the first year of analysis data in each grid cell as the climatology. This will in most cases be the true climatology, as very few cells have observational influence in the first year. An example of this replacement process for $w_0 = 0.5$ can be seen in Supplementary Fig. 7. Supplementary Fig. 15 shows how this modification shifts the global distribution of autocorrelation to higher values.

## Data availability

The HadISST1, HadSST4, HadCRUT5 and EN4.2.2 datasets used in this study are all available at https://www.metoffice.gov.uk/hadobs/. The ERSSTv5 operational dataset used in this study is available at https://psl.noaa.gov/data/gridded/data.noaa.ersst.v5.html, and the ERSSTv5 uncertainties are available at https://www.ncei.noaa.gov/pub/data/cmb/ersst/v5/ensemble.1854-2017/. No new data has been produced.

## Code availability

All code used to analyse the data and generate figures can be accessed at https://github.com/mayaby/AMOC_EWS_uncertainties[51].

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

## Acknowledgements

M.B.Y. and N.B. acknowledge funding by the European Union's Horizon 2020 research and innovation programme under the Marie Sklodowska-Curie grant agreement No.956170. N.B. and S.B. acknowledge funding by the Volkswagen foundation. This is TiPES contribution # 256; the TiPES ('Tipping Points in the Earth System') project has received funding from the European Union's Horizon 2020 research and innovation programme under grant agreement No. 820970. NB acknowledges further funding by the German Federal Ministry of Education and Research under grant No. 01LS2001A. We thank John Kennedy for helpful discussions.

## Author contributions

M.B.Y., V.S. and N.B. conceived the study and designed it with contributions from S.B. M.B.Y. carried out the analysis. M.B.Y., V.S., S.B. and N.B. discussed results, and M.B.Y. wrote the paper with contributions from V.S., S.B. and N.B.

## Funding

## Competing interests

The authors declare no competing interests.
