## [Peer Review File · Nature Communications]

Uncertainties in critical slowing down indicators of observation-based fingerprints of the Atlantic Overturning CirculationREVIEWER COMMENTS

Reviewer #1 (Remarks to the Author):

The manuscript entitled “Uncertainties in critical slowing down indicators of observation-based fingerprints of the Atlantic Overturning Circulation”, discussed the biases related to changes in available observations with time for sea surface temperature and salinity datasets when quantifying the critical slowing down (CSD) indicators for the AMOC, and provided conservative surrogate-based significance testing methods to address these issues. It is shown that with the approach proposed by the authors, the regions with significant CSD are reduced and are more accurately reflecting the AMOC strength compared with previous studies.

The study provided an approach to quantify and resolve the uncertainties caused by pre-processing of the observational data, including gap filling. It contributes to accurate and more robust estimation of the destabilization of a system from the observations. Therefore, I recommend the manuscript for publication given that the comments below are sufficiently addressed.

Introduction- are alternative approaches to quantify uncertainties related to non-stationary observations used in previous studies on CSD of AMOC, e.g., in Boers (2021), Rahmstorf et al., 2015, Caesar et al., 2018? There are discussions on methods to calculate the CSD indicators, but a review on ways to address the uncertainties of the observational data is absent.

Line 91- can you define “full grid cells”? Does it refer to grid cells that have no missing data? It would be helpful to clarify.

Line 97- “sampling uncertainty around each bias ensemble member,” it is unclear what does this mean. Could you clarify?

Figure 1- the colors for SPG and global seem to be different in the figure and in the caption.

Minor typos:

Line 21- “these include the the Greenland Ice Sheet...”

Reviewer #2 (Remarks to the Author):

General comments:

This study aims to assess how uncertainties in observational datasets affect the detection of critical slowing down (CSD) indicators of the Atlantic Meridional Overturning Circulation (AMOC). The authors

found that the uncertainties introduced by missing data imputation do contribute to the uncertainties in CSD indicators; however, the AMOC becoming increasingly unstable remains a significant signal based on AMOC fingerprints. While this is an important topic, I have several major comments for the authors to consider.

Specific comments

1. The authors used SST and SSS-based fingerprint index to infer AMOC. However, several recent studies have shown that Subpolar North Atlantic SSTA is not a good indicator of AMOC variability (e.g., Little et al. 2020; Zhu et al. 2023). While I agree with the shortage of AMOC observations to determine its long-term changes, I'm wondering whether the climate models overall suggest the same change towards a more unstable AMOC in the past century.

References:

Little, C. M., Zhao, M., & Buckley, M. W. (2020). Do surface temperature indices reflect centennial-timescale trends in Atlantic Meridional Overturning Circulation strength? *Geophysical Research Letters*, 47, e2020GL090888.

Zhu, C., Liu, Z., Zhang, S., & Wu, L. (2023). Likely accelerated weakening of Atlantic overturning circulation emerges in optimal salinity fingerprint. *Nature Communications*, 14, 1245.

2. The study uses λ to quantify the stability of AMOC based on SSTA and SSSA. However, λ could come from multiple processes, such as air-sea heat flux feedback and oceanic damping via vertical entrainment. These processes have no direct association with AMOC. Thus, I don't think λ trend reflects AMOC stability change.

3. From Figure 3 and Figure 4, the significant λ trend is located mostly southeastward of the Iceland, which is not really over the subregions of the subpolar North Atlantic used to define the SST-based AMOC fingerprint index (i.e., the Labrador Sea and Irminger Sea).

4. Only HadISST1 and HadCRUT are considered in this study. With multiple century-long SST datasets available, the authors can further test their results by incorporating these different datasets. To my knowledge, ERSSTv4 and ERSSTv5 both provides ensemble members whose SST are calculated by perturbing the parameters during the SST reconstruction.

RESPONSE TO REVIEWER COMMENTS

Please find our point-by-point responses to the concerns raised by the two referees in blue font below.

Reviewer 1

Dear Editor and Authors,

The manuscript entitled “Uncertainties in critical slowing down indicators of observation-based fingerprints of the Atlantic Overturning Circulation”, discussed the biases related to changes in available observations with time for sea surface temperature and salinity datasets when quantifying the critical slowing down (CSD) indicators for the AMOC, and provided conservative surrogate-based significance testing methods to address these issues. It is shown that with the approach proposed by the authors, the regions with significant CSD are reduced and are more accurately reflecting the AMOC strength compared with previous studies.

The study provided an approach to quantify and resolve the uncertainties caused by pre-processing of the observational data, including gap filling. It contributes to accurate and more robust estimation of the destabilization of a system from the observations. Therefore, I recommend the manuscript for publication given that the comments below are sufficiently addressed.

Thank you very much for the accurate summary of our study and the overall positive evaluation. Please find point-by-point responses to your concerns, as well as references to how we revised our manuscript, in the following. Your suggestions have helped us improve our manuscript.

Introduction- are alternative approaches to quantify uncertainties related to non-stationary observations used in previous studies on CSD of AMOC, e.g., in Boers (2021), Rahmstorf et al., 2015, Caesar et al., 2018? There are discussions on methods to calculate the CSD indicators, but a review on ways to address the uncertainties of the observational data is absent.

Thank you for this comment. In previous work the uncertainties with respect to different ways to detrend the time series, as well as with respect to the size of the window length have been presented; see e.g. Boers and Rypdal (2021) or Boers (2021). However, as far as we are aware, no other study has addressed uncertainties in CSD indicators stemming from the actual underlying data. We have highlighted in line 37-8 of the revised introduction that our paper is the first to address this issue.

Line 91- can you define “full grid cells”? Does it refer to grid cells that have no missing data? It would be helpful to clarify.

We agree that this wording is not clear. Yes, “full grid cells” refer to grid cells that are not empty - in HadCRUT and HadSST, there are some months for which some of the cells have no data (i.e. are set to “nan”). We have clarified this in the relevant sections of the revised manuscript (line 95).

Line 97- “sampling uncertainty around each bias ensemble member,” it is unclear what does this mean. Could you clarify?

We agree this merits further explanation. HadSST4 provides three types of uncertainties (metoffice.gov.uk):

- “Measurement uncertainty” - a value for the uncertainty associated with the measurement error

- “Sampling uncertainty” - a value for the uncertainty associated with under-sampling
- “Bias ensemble” - 200 ensemble members in which the different parameters associated with the bias correction have been varied

The sampling uncertainty is given in equation 5 of Kennedy et al. (2019). It accounts for the fact that when there are more observations, the accuracy of the SST value in that gridcell is higher. The sampling uncertainty thus scales as the inverse of the number of so-called superobservations in a given grid cell. These superobservations are calculated in the initial step of the gridding, when the raw measurements are sorted into 1° latitude by 1° longitude by pentad bins. The superobservations are later sorted and averaged onto the final 5° latitude by 5° longitude grid.

We added a sentence in the manuscript to refer to the methods, where we have further clarified this point (lines 102-3).

Figure 1- the colors for SPG and global seem to be different in the figure and in the caption.

Thank for this careful observation, we have corrected this typo in the caption.

Minor typos: Line 21- “these include the the Greenland Ice Sheet...”

We have corrected this typo.

Reviewer 2

General comments: This study aims to assess how uncertainties in observational datasets affect the detection of critical slowing down (CSD) indicators of the Atlantic Meridional Overturning Circulation (AMOC). The authors found that the uncertainties introduced by missing data imputation do contribute to the uncertainties in CSD indicators; however, the AMOC becoming increasingly unstable remains a significant signal based on AMOC fingerprints. While this is an important topic, I have several major comments for the authors to consider.

Thank you for taking the time to thoroughly review our paper. Your comments and suggestions were very helpful for us when revising and improving our manuscript. Please find detailed-point-by-point responses in the following.

Specific comments 1. The authors used SST and SSS-based fingerprint index to infer AMOC. However, several recent studies have shown that Subpolar North Atlantic SSTA is not a good indicator of AMOC variability (e.g., Little et al. 2020; Zhu et al. 2023). While I agree with the shortage of AMOC observations to determine its long-term changes, I’m wondering whether the climate models overall suggest the same change towards a more unstable AMOC in the past century.

References: Little, C. M., Zhao, M., Buckley, M. W. (2020). Do surface temperature indices reflect centennial-timescale trends in Atlantic Meridional Overturning Circulation strength? *Geophysical Research Letters*, 47, e2020GL090888. Zhu, C., Liu, Z., Zhang, S., Wu, L. (2023). Likely accelerated weakening of Atlantic overturning circulation emerges in optimal salinity fingerprint. *Nature Communications*, 14, 1245.

We agree that this is an important point. The validity of different AMOC indicators is still debated, and we address this point in our discussion. Essentially the uncertainties we discuss in our study are twofold, namely (i) regarding the effects of measurement uncertainties in the underlying observational datasets on CSD indicators and

how to thoroughly quantify and propagate them, and (ii) regarding the interpretation of the fingerprints derived from these datasets, and of their stability changes, in terms of how well the different fingerprints represent the AMOC. We thus fully acknowledge the caveats associated with the different AMOC fingerprints proposed in the literature. Given these caveats, we consider the best approach to study the stability of the AMOC is to use a large number of fingerprints, both SST and salinity based, as was done in Boers 2021. Our study focuses on quantifying and propagating the effects of the uncertainties in several SST and salinity datasets where possible, but we have also substantially expanded our discussion section and now put more emphasis on the uncertainty related to the representativeness of different fingerprints for the AMOC and its stability changes.

The arguments in the literature against using Subpolar North Atlantic SSTA as an AMOC indicator are usually focused on whether or not the mean state of the AMOC has been weakening in the historical period (Latif et al., 2014; Zhu et al., 2023; Kilbourne et al., 2022). The detection of CSD is not connected to changes in the mean state, and thus these arguments usually do not apply to CSD. For example, the SSTA fingerprint has been criticized for containing more noise than other fingerprints (Zhu et al., 2023), for being too sensitive to other forcings such as aerosols (Little et al., 2020) and for its mean trend depending on the forcing scenario (Menary et al., 2020) - none of which make it less conducive to a CSD analysis. It is also true that the North Atlantic SSTA does not seem to correlate well with the strength of the AMOC at 26N on annual timescales, but this is in fact to be expected, as they represent different parts of the AMOC (Jackson et al., 2022; Gu et al., 2020). This is reflected in the studies that do find longer term or lagged correlations for the SSTA fingerprint (Jackson and Wood, 2020; Zhu et al., 2023). A change in stability in the AMOC system should be evident in all parts of the system, and so the focus on its subpolar component does not invalidate the CSD analysis.

Finally, there is also the issue that while the North Atlantic SSTA is correlated to some extent with the AMOC at 26N, the extent of this correlation varies for different forcing scenarios and time periods (Little et al., 2020; Jackson and Wood, 2020). This is also reflected in the studies that find that, while the North Atlantic warming hole is mainly caused by changes in ocean transport, atmospheric processes also play a part in changing the SSTs in that region (Drijfhout et al., 2012; Menary and Wood, 2018; Li et al., 2022; He et al., 2022). Therefore, if one is interested in capturing the short-term variability of the AMOC, the SSTA fingerprint might not be ideal. However, using this fingerprint for CSD analysis does not require that the overall variability of the fingerprint is the same as that of the AMOC. What we are interested in is the ability of the AMOC to restore from perturbations. One of the main identified “restoring forces” of sub-polar gyre (SPG) perturbations is the feedback cycle in which heat loss leads to increased deep water formation, which in turns leads to a stronger circulation and more heat transport into the SPG (Menary et al., 2015; Sun et al., 2021). This “restoring force” is intrinsically connected to the AMOC - if the AMOC becomes less stable, it takes longer for the ocean circulation transporting heat into the SPG to restore from perturbations, and thus the SPG becomes less stable. It thus doesn't matter if the perturbations to the SPG are different from the perturbations to the AMOC streamfunction (i.e. the variability is different), because in theory their ability to restore from these differing perturbations changes together.

We should note, however, that this only holds when we use CSD indicators that allow for the underlying noise to be non-stationary (such as λ in our work). The variance, for example, would be strongly influenced if the source of SPG variability changes, because the amplitude of the variability could change.

In addition to the question of what is a good fingerprint of the AMOC, we also agree with the referee that it is interesting to investigate how the AMOC stability changed in historical CMIP simulations. However, it is difficult to interpret these results, as the AMOC is suspected to be “too stable” in climate models (Mecking et al., 2017; Liu et al., 2017), and even if it does destabilize it may only do so later in the 21st century (Romanou et al.). We are currently working on an in-depth analysis of the CMIP6 archive. We have made changes to the revised manuscript to highlight this uncertainty in the connection of the indicators to the AMOC. Also, we have substantially expanded our discussion to incorporate the above thoughts, following your suggestions (lines 215-259).

2. The study uses λ to quantify the stability of AMOC based on SSTA and SSSA. However, λ could come from multiple processes, such as air-sea heat flux feedback and oceanic damping via vertical entrainment. These processes have no direct association with AMOC. Thus, I don't think λ trend reflects AMOC stability change.

This is a key challenge with CSD indicators, as the reviewer points out. In particular for the more classical indicators of CSD, namely the variance and the lag-one autocorrelation, it has recently been shown that they can increase for reasons different from stability loss and would thus give false alarms (Boers, 2021; Boettner and Boers, 2021; Smith et al., 2023). However, it has also been shown that the restoring rate λ as estimated here is more robust than the classical indicators (Boers, 2021).

Specifically regarding the referee's concern that other processes might lead to a λ increase: Processes that act on shorter time-scales, such as air-sea heat fluxes, act as the perturbing noise, and λ is the restoring rate of the response to those perturbations. When we calculate λ , we account for the fact that the variance and autocorrelation of the external forcing noise could change over time. This is in fact the strength of using λ over the variance and autocorrelation, where changes in the noise can bias the results. The increase in λ in a given ocean region indicates that that region has become slower at restoring from perturbations. If the AMOC were destabilizing, this is what we would expect to happen - increasing recovery times over the AMOC transport pathways.

Of course, as the reviewer rightly points out, this does not preclude that the slowing down that we see is due to a long-term change in another internal ocean process. For example, a gradual increase in mixed layer depth would increase the effective heat capacity and with it the response time. However, we consider it implausible that this effect explains the long-term trend in λ , as the ocean mixed layer is not observed to be deepening overall and global warming is in fact expected to increase the stratification of the upper ocean (Shi et al., 2022). In addition, such damping of fluctuations would not only increase λ but also decrease the variance, and we find that in these regions the variance has actually been increasing in the most recent decades, as one would expect from CSD (Fig S2, S10).

We note that, for robustness, one should not rely on a single CSD indicator, but compare several and we have hence calculated the variance and autocorrelation for all the datasets for comparison (see SI figures).

We would like to emphasize that, as noted above, in our study we address the twofold uncertainties in CSD analyses of previously proposed AMOC fingerprints, namely (i) regarding the effect of measurement uncertainties of the underlying datasets, for which we develop a comprehensive methodology, and (ii) regarding the interpretation of the fingerprints in terms of AMOC proxies. We do apply our methods to indices that have previously been demonstrated, based on models, to be (to varying extent) representative of the high-northern latitude AMOC (Drijfhout et al., 2012; Rahmstorf et al., 2015;

Caesar et al., 2018; Jackson and Wood, 2020; Jackson et al., 2022). However, we fully acknowledge the uncertainties associated with interpreting the fingerprints as AMOC proxies and discuss this in detail in our manuscript; see e.g. lines 215-230. As stated above, we have made changes to the text to further emphasize the uncertainties related to interpreting the increases in λ in terms of a stability decline of the AMOC, including the discussion of different processes that also affect the restoring rate λ (lines 231-242).

3. From Figure 3 and Figure 4, the significant λ trend is located mostly southeastward of the Iceland, which is not really over the subregions of the subpolar North Atlantic used to define the SST-based AMOC fingerprint index (i.e., the Labrador Sea and Irminger Sea).

This is correct, and we thank the reviewer for pointing it out. We discuss this in the final section of the paper, but realize that the difference between the fingerprint and the global maps was not explained adequately.

However, we would like to note that the increase in λ in the Labrador and Irminger seas is still present in most grid cells, but is simply not significant on an individual grid cell level. This may seem to contradict the significant increase found in the SST-based AMOC fingerprint (Fig 2), but in fact when averaging the SST over an extended region one gets rid of local processes and improves the observational accuracy, allowing for the overall trend in λ to become significant.

This is the strength of combining both averaged indices and geographic cell-by-cell analyses. The maps do reveal, as the reviewer rightly points out, that the λ is more significant along the North Atlantic Current and in the GIN sea than it is in the Labrador sea or the sub-polar gyre (SPG). This is actually in line with recent results from the OS-NAP array that deep water formation in the Labrador sea is less important for the AMOC than deep water from the GIN and Irminger seas (Lozier et al., 2019). It should be noted though that those results do not invalidate the use of the SPG based index for studying the AMOC, as the SPG still plays an important role in the AMOC through e.g. transformation of water masses in the Irminger sea (Swingedouw et al., 2022). However, this more secondary role in the circulation could explain why trends in λ are less significant in the SPG.

We have made changes to the revised manuscript to highlight this difference in regions in lines 243-259.

4. Only HadISST1 and HadCRUT are considered in this study. With multiple century-long SST datasets available, the authors can further test their results by incorporating these different datasets. To my knowledge, ERSSTv4 and ERSSTv5 both provide ensemble members whose SST are calculated by perturbing the parameters during the SST reconstruction.

Thank you very much for the suggestion to expand our analysis to the ERSST data! Although there are many available SST datasets, as the focus of our manuscript is the calculation of uncertainties, we only incorporate either datasets that have uncertainty estimates (HadSST4, HadCRUT5), or those that are used by Boers 2021 (EN4, HadISST). We hence thank the author for pointing out that the ERSST has an uncertainty ensemble. We have repeated our analysis for ERSSTv5, and have added the results to the figures and to the appropriate sections in the revised manuscript (Figs. 2-4,6,S1-4,S9-10,S13-14). As far as we are aware, there are no other historical SST datasets with publicly available uncertainty estimates.

REFERENCES

- N. Boers. Observation-based early-warning signals for a collapse of the Atlantic Meridional Overturning Circulation. *Nature Climate Change*, 11(8):680–688, 2021. ISSN 17586798. doi: 10.1038/s41558-021-01097-4. URL <http://dx.doi.org/10.1038/s41558-021-01097-4>.
- N. Boers and M. Rypdal. Critical slowing down suggests that the western Greenland Ice Sheet is close to a tipping point. *Proceedings of the National Academy of Sciences of the United States of America*, 118(21):1–7, 2021. ISSN 10916490. doi: 10.1073/pnas.2024192118.
- C. Boettner and N. Boers. Critical Slowing Down In Dynamical Systems Driven By Non-Stationary Correlated Noise. (1):1–7, 2021.
- L. Caesar, S. Rahmstorf, A. Robinson, G. Feulner, and V. Saba. Observed fingerprint of a weakening Atlantic Ocean overturning circulation. *Nature*, 556(7700):191–196, 2018. ISSN 14764687. doi: 10.1038/s41586-018-0006-5.
- S. Drijfhout, G. J. van Oldenborgh, and A. Cimadoribus. Is a decline of AMOC causing the warming hole above the North Atlantic in observed and modeled warming patterns? *Journal of Climate*, 25(24):8373–8379, 2012. ISSN 08948755. doi: 10.1175/JCLI-D-12-00490.1.
- S. Gu, Z. Liu, and L. Wu. Time Scale Dependence of the Meridional Coherence of the Atlantic Meridional Overturning Circulation. *Journal of Geophysical Research: Oceans*, 125(3):1–10, 2020. ISSN 21699291. doi: 10.1029/2019JC015838.
- C. He, A. C. Clement, M. A. Cane, L. N. Murphy, J. M. Klavans, and T. M. Fenske. A North Atlantic Warming Hole Without Ocean Circulation. *Geophysical Research Letters*, 49(19), 2022. ISSN 19448007. doi: 10.1029/2022GL100420.
- L. C. Jackson and R. A. Wood. Fingerprints for early detection of changes in the AMOC. *Journal of Climate*, 33(16):7027–7044, 2020. ISSN 08948755. doi: 10.1175/JCLI-D-20-0034.1.
- L. C. Jackson, A. Biastoch, M. W. Buckley, D. G. Desbruyères, E. Frajka-Williams, B. Moat, and J. Robson. The evolution of the North Atlantic Meridional Overturning Circulation since 1980. *Nature Reviews Earth and Environment*, 0123456789, 2022. ISSN 2662138X. doi: 10.1038/s43017-022-00263-2.
- J. J. Kennedy, N. A. Rayner, C. P. Atkinson, and R. E. Killick. An Ensemble Data Set of Sea Surface Temperature Change From 1850: The Met Office Hadley Centre HadSST.4.0.0.0 Data Set. *Journal of Geophysical Research: Atmospheres*, 124(14): 7719–7763, 2019. ISSN 21698996. doi: 10.1029/2018JD029867.
- K. H. Kilbourne, A. D. Wanamaker, P. Moffa-Sanchez, D. J. Reynolds, D. E. Amrhein, P. G. Butler, G. Gebbie, M. Goes, M. F. Jansen, C. M. Little, M. Mette, E. Moreno-Chamarro, P. Ortega, B. L. Otto-Bliesner, T. Rossby, J. Scourse, and N. M. Whitney. Atlantic circulation change still uncertain. *Nature Geoscience*, pages 13–16, 2022. ISSN 1752-0894. doi: 10.1038/s41561-022-00896-4.
- M. Latif, J. Sun, M. Visbeck, M. H. Bordbar, M. Hadi Bordbar, and M. H. Bordbar. Natural variability has dominated Atlantic Meridional Overturning Circulation since 1900. *Nature Climate Change*, 12(5):455–460, 2014. ISSN 17586798. doi: 10.1038/s41558-022-01342-4.
- L. Li, M. S. Lozier, and F. Li. Century-long cooling trend in subpolar North Atlantic forced by atmosphere: an alternative explanation. *Climate Dynamics*, 58(9-10): 2249–2267, 2022. ISSN 14320894. doi: 10.1007/s00382-021-06003-4. URL <https://doi.org/10.1007/s00382-021-06003-4>.
- C. M. Little, M. Zhao, and M. W. Buckley. Do Surface Temperature Indices Reflect Centennial-Timescale Trends in Atlantic Meridional Overturning Circulation

- Strength? *Geophysical Research Letters*, 47(22):1–10, 2020. ISSN 19448007. doi: 10.1029/2020GL090888.
- W. Liu, S. P. Xie, Z. Liu, and J. Zhu. Overlooked possibility of a collapsed atlantic meridional overturning circulation in warming climate. *Science Advances*, 3(1):1–8, 2017. ISSN 23752548. doi: 10.1126/sciadv.1601666.
- M. S. Lozier, F. Li, S. Bacon, F. Bahr, A. S. Bower, S. A. Cunningham, M. F. De Jong, L. De Steur, B. DeYoung, J. Fischer, S. F. Gary, B. J. Greenan, N. P. Holliday, A. Houk, L. Houpert, M. E. Inall, W. E. Johns, H. L. Johnson, C. Johnson, J. Karstensen, G. Koman, I. A. Le Bras, X. Lin, N. Mackay, D. P. Marshall, H. Mercier, M. Oltmanns, R. S. Pickart, A. L. Ramsey, D. Rayner, F. Straneo, V. Thierry, D. J. Torres, R. G. Williams, C. Wilson, J. Yang, I. Yashayaev, and J. Zhao. A sea change in our view of overturning in the sub-polar North Atlantic. *Science*, 363(6426):516–521, 2019. ISSN 10959203. doi: 10.1126/science.aau6592.
- J. V. Mecking, S. S. Drijfhout, L. C. Jackson, and M. B. Andrews. The effect of model bias on Atlantic freshwater transport and implications for AMOC bi-stability. *Tellus, Series A: Dynamic Meteorology and Oceanography*, 69(1):1–15, 2017. ISSN 16000870. doi: 10.1080/16000870.2017.1299910. URL <http://dx.doi.org/10.1080/16000870.2017.1299910>.
- M. B. Menary and R. A. Wood. An anatomy of the projected North Atlantic warming hole in CMIP5 models. *Climate Dynamics*, 50(7-8):3063–3080, 2018. ISSN 14320894. doi: 10.1007/s00382-017-3793-8.
- M. B. Menary, D. L. Hodson, J. I. Robson, R. T. Sutton, and R. A. Wood. A mechanism of internal decadal Atlantic Ocean variability in a high-resolution coupled climate model. *Journal of Climate*, 28(19):7764–7785, 2015. ISSN 08948755. doi: 10.1175/JCLI-D-15-0106.1.
- M. B. Menary, J. Robson, R. P. Allan, B. B. Booth, C. Cassou, G. Gastineau, J. Gregory, D. Hodson, C. Jones, J. Mignot, M. Ringer, R. Sutton, L. Wilcox, and R. Zhang. Aerosol-Forced AMOC Changes in CMIP6 Historical Simulations. *Geophysical Research Letters*, 47(14), 2020. ISSN 19448007. doi: 10.1029/2020GL088166.
- S. Rahmstorf, J. E. Box, G. Feulner, M. E. Mann, A. Robinson, S. Rutherford, and E. J. Schaffernicht. Exceptional twentieth-century slowdown in Atlantic Ocean overturning circulation. *Nature Climate Change*, 5(5):475–480, 2015. ISSN 17586798. doi: 10.1038/nclimate2554.
- A. Romanou, D. Rind, J. Jonas, R. Miller, M. Kelley, G. Russell, C. Orbe, L. Nazarenko, R. Latta, and G. A. Schmidt. Stochastic Bifurcation of the North Atlantic Circulation Under A Mid-Range Future Climate Scenario With The NASA-GISS ModelE.
- H. Shi, F. F. Jin, R. C. Wills, M. G. Jacox, D. J. Amaya, B. A. Black, R. R. Rykaczewski, S. J. Bograd, M. García-Reyes, and W. J. Sydeman. Global decline in ocean memory over the 21st century. *Science Advances*, 8(18):1–10, 2022. ISSN 23752548. doi: 10.1126/sciadv.abm3468.
- T. Smith, R. M. Zotta, C. A. Boulton, T. M. Lenton, W. Dorigo, and N. Boers. Reliability of resilience estimation based on multi-instrument time series. *Earth System Dynamics*, 14(1):173–183, 2023. ISSN 21904987. doi: 10.5194/esd-14-173-2023.
- J. Sun, M. Latif, and W. Park. Subpolar gyre–AMOC–atmosphere interactions on multi-decadal timescales in a version of the Kiel climate model. *Journal of Climate*, 34(16): 6583–6602, 2021. ISSN 15200442. doi: 10.1175/JCLI-D-20-0725.1.
- D. Swingedouw, M. N. Houssais, C. Herbaut, A. C. Blaizot, M. Devilliers, and J. Deshayes. AMOC Recent and Future Trends: A Crucial Role for Oceanic Resonance and Greenland Melting? *Frontiers in Climate*, 4(April), 2022. ISSN 26249553. doi: 10.3389/fclim.2022.838310.
- C. Zhu, Z. Liu, S. Zhang, and L. Wu. Likely accelerated weakening of Atlantic overturn-

ing circulation emerges in optimal salinity fingerprint. *Nature Communications*, 14 (1):1–9, 2023. ISSN 20411723. doi: 10.1038/s41467-023-36288-4.

REVIEWERS' COMMENTS

Reviewer #1 (Remarks to the Author):

The authors have addressed my comments and have modified the manuscript accordingly. The newly added discussion on the uncertainties regarding the representativeness of the CSD indicators and discussion on other potential processes that might be affecting these indicators besides the AMOC are informative. The additional SST dataset incorporated in the revision added confidence and comprehensiveness to the findings. Therefore I recommend the manuscript for publication.